# Detection and Quantification of Botanical Impurities in Commercial Oregano (*Origanum vulgare*) Using Metabarcoding and Digital PCR

**DOI:** 10.3390/foods12162998

**Published:** 2023-08-09

**Authors:** Antoon Lievens, Valentina Paracchini, Linda Garlant, Danilo Pietretti, Alain Maquet, Franz Ulberth

**Affiliations:** 1European Commission, Joint Research Centre (JRC), B-2440 Geel, Belgium; 2European Commission, Joint Research Centre (JRC), I-21027 Ispra, Italy

**Keywords:** oregano, food authentication, ddPCR, NGS, metabarcoding

## Abstract

DNA technology for food authentication is already well established, and with the advent of Next Generation Sequencing (NGS) and, more specifically, metabarcoding, compositional analysis of food at the molecular level has rapidly gained popularity. This has led to several reports in the media about the presence of foreign, non-declared species in several food commodities. As herbs and spices are attractive targets for fraudulent manipulation, a combination of digital PCR and metabarcoding by NGS was employed to check the purity of 285 oregano samples taken from the European market. By using novel primers and analytical approaches, it was possible to detect and quantify both adulterants and contaminants in these samples. The results highlight the high potential of NGS for compositional analysis, although its quantitative information (read count percentages) is unreliable, and other techniques are therefore needed to complement the sequencing information for assessing authenticity (‘true to the name’) of food ingredients.

## 1. Introduction

Oregano (*Origanum vulgare*) is a herb native to temperate Western and Southwestern Eurasia and the Mediterranean region. Its use as a herb dates back to ancient times [1], and today it is widely used in Mediterranean cuisine, the Philippines, and Latin America, especially in Argentina. In addition, oregano possesses anti-oxidative properties, and is also used by food processors to prevent or minimize rancidity in foods with high fat content [2]. The dried herb, tincture, and its essential oil have applications as food flavorings, and are used in certain liqueur formulations as well.

Marketing standards for herbs and spices, including oregano, do not exist in the legal framework of the EU. However, voluntary standards, such as those developed by the International Organization for Standardization (ISO), are used by trading partners to control quality of goods. ISO 7925:1999 [3] sets provisions for certain parameters inter alia for extraneous matter, i.e., all that does not belong to the leaves of oregano (*Origanum* genus, species, and sub-species, excluding *O. majorana*) and all other extraneous matter of animal, vegetable, and mineral origin. The total percentage of extraneous matter, determined according to ISO 927:2009 [4], which is based on visual inspection, should not be more than 1% (by mass) for processed and 3% (by mass) for semi-processed oregano.

Once oregano is processed (cut, crushed, milled), extraneous matter, including the presence of non-declared botanicals, is difficult to determine visually. Therefore, physico-chemical methods have been used to assess the purity of oregano, e.g., NMR [5] and IR spectroscopy [6] and liquid chromatography coupled with mass spectrometry (LC-MS) [7]. In the latter publication, Black et al. [7] tested 53 samples taken from the markets of the UK and Ireland and 25 samples purchased from the Internet, and found 24% of samples adulterated with olive or myrtle leaves at levels between 30 and 70% using a tiered approach involving IR spectroscopy and LC-MS.

The use of DNA technology for food authentication [8,9,10], and specifically for plant material [11,12], has been well established. In particular, Next Generation Sequencing (NGS), especially metabarcoding analysis, is increasingly employed in food and nutrition science for various purposes [13,14,15,16]. Its use for compositional analysis to detect the presence of foreign botanicals in herbs and spices has been reported in the scientific literature [17,18], as well as in the media. One study found 82% of the oregano samples tested to contain another plant species, and half of the oregano samples tested contained bindweed (*Convolvulus arvensis*), a potentially toxic common weed [19]. Although the technique is becoming more widely available, its successful implementation requires access to dedicated instrumentation, bioinformatics pipelines, and experienced operators. In addition, interpretation of results can be challenging and misleading, especially regarding the percentage distribution of sequencing reads across the species reported.

To create a more in-depth understanding of the opportunities and limitations of DNA-based techniques for assessing the authenticity of plant material, we conducted a market survey by analyzing 285 commercial oregano samples collected from 20 EU member states. Samples were analyzed with metabarcoding by NGS for their compositional analysis and droplet digital PCR (ddPCR) for their quantification. We show that, while NGS provides exhaustive results in terms of compositional analysis (presence of species in a sample), the quantitative information (read counts) cannot be relied on without traditional quantitative DNA techniques to complement the sequencing information. In addition, several primer sets for the semi-quantitative determination of adulterants in oregano are introduced as well.

## 2. Materials and Methods

### 2.1. Samples

**Reference samples:** Plant reference material from the living outdoor collection was obtained from the Meise Botanical Garden, (Meise, Belgium) (one sample of *Convolvulus arvensis* 19630417, *Olea europaea* 19084269, and *Myrtus communis* 20040425-31, three samples of *Origanum vulgare* 19580630, 19840672, and 20080080-13). Reference plant DNA (two samples of *Chenopodium album* DB 2897 and DB 1717, one sample of *C. arvensis* DB 13466, *Cistus incanus* 19237, *O. europaea* 11214, *O. vulgare hirtum* DB 2760, *O. vulgare* DB 5739, and *M. communis* 10348) was obtained from the Kew DNA bank (The Royal Botanic Gardens, Kew, UK. https://www.kew.org/data/dnaBank/ (accessed on 1 July 2019)) and from the DNA bank of the Botanic Garden and Botanical Museum Berlin (Germany). All DNA samples, as well as underlying voucher specimens, are deposited at the Botanic Garden and Botanical Museum Berlin, and are available via the Global Genome Biodiversity Network [20] and the Global Biodiversity Information Facility. Plant material and DNA were provided under the agreement of the Convention on Biological Diversity 1992.

**Samples used for method development and in-house validation:** Samples were prepared from fresh and dry materials commercially available through shops and garden centers (one sample of *C. album*, *M. communis*, *Thymus vulgaris*), or from the above mentioned collections. Where possible, single plants/fruits were used.

The European Spice Association (ESA, Reuterstraße 151, 53113 Bonn, Germany) provided an oregano quality control sample, which, in addition to oregano, contained olive (*O. europaea*), myrtle (*M. communis*), cistus (*C. incanus*), sumac (*Rhus* spp.), hazel (*Corylus* spp.), thyme (*Thymus vulgaris*), and bindweed (*C. arvensis*) in the following proportions: *Origanum vulgare* 22%, *Origanum onites* 22%, olive leaf 10%, myrtle leaf 10%, cistus leaf 10%, sumac leaf 10%, hazel leaf 10%, thyme 5%, bindweed 1% (for more information on this sample, see [21]).

**Commercial oregano samples:** Samples (*n* = 285) were collected from 20 EU member states at various stages of the supply chain and processing (i.e., whole, crushed, and ground).

**Sample preparation:** Fresh materials were thinly sliced (<1 mm) and dried (Memmert model UM500 at 75 °C) for 2 h or until dry. Dry and dried materials were mixed/milled using an MM301 ball mill (Retsch) for 2 min at 30 Hz using either 10 mL grinding jars and 10 mm beads (for softer and pre-ground materials) or 20 mL grinding jars and 20 mm beads (for harder materials, such as seeds). Sample blends were prepared from single species, and their mixing ratio (weight/weight percentage) was determined gravimetrically (Mettler Toledo PG503-S).

### 2.2. DNA Extraction

**Extraction protocol:** Automated DNA extraction of approximately 300 mg plant material was performed using a Tecan Freedom EVO liquid handler with Promega chemicals (CTAB extraction buffer, CLD lysis buffer, Reliaprep Resin, BWA wash buffer), the Promega Purefood protocol, a sample load volume of 350 μL, and an elution volume of 150 μL. Large-volume DNA extraction (>350 mg sample) was performed manually using a CTAB-based method adopted from [22].

**DNA quantification:** Fluorometric DNA quantification was performed on a Qubit 4 fluorometer (Invitrogen, Merelbeke, Belgium) with high-sensitivity chemistry (Invitrogen), according to manufacturer instructions, using 5 μL of sample. For each sample, two independent sample dilutions were quantified twice (two independent standard curves), thus yielding four measurements per sample, which were averaged to obtain the DNA concentration estimate.

### 2.3. PCR Primers

In this study, most species-specific primers were used with intercalating dyes (SYBRgreen or EVAgreen) in both qPCR and ddPCR, with the exception of Oregano/Olive, for which a ddPCR duplex assay was developed. For the other assays, probe sequences were designed as well, but were not validated (probe sequences and validation status are given in Table A2 of Appendix A).

Oregano *(Origanum vulgare)* primers (Orvu EF1 F/R) target the gene for Elongation factor 1, and were adopted from [23]. A matching probe was developed in house.Olive tree *(Olea europaea)*-specific primers (Oleu SADbis F/R) were designed in house and target the Stearoyl-acyl carrier protein desaturase gene (SAD1), which is associated with the oleic acid composition of olives [24].White goosefoot *(Chenopodium album)* primers (Chal pc-1E1p1 F/R) were designed in house and target the phosphoenolpyruvate carboxylase (ppc-1E1) gene (KJ161681.1), a key enzyme of both the CAM and C4 pathways [25].Bindweed *(Convolvulus arvensis)* primers (Coar HSSp2 F/R) were designed in house and target the phi1 Homospermidine synthase (HSS) pseudogene (HF911513.1) [26].Myrtle *(Myrtus communis)* primers (MyrtusP1 F/R) were designed in house and target an isoprene synthase gene.Cistus *(Cistus incanus)* primers (Cistus S13593 F/R) were designed in house and target a geranylgeranyl pyrophosphate synthase (GGPPS1) gene.

**Specificity:** Primer pairs were tested for cross-reactivity with other species using qPCR and SYBRgreen chemistry. For the results, see Table A3 in Appendix A.

### 2.4. PCR Methods

**Real-time PCR** reactions were performed in 25 μL using primers from Table 1 ordered from Invitrogen (standard desalted primers). Reactions were run using the Powerup SYBRgreen mastermix (Life Technologies, Merelbeke, Belgium) and nuclease-free water (Ambion, Huntingdon, UK). Final primer concentration was 200 nM. DNA template input was 18 ng per reaction, unless otherwise mentioned. All reactions were amplified in ABI microamp 96-well 0.1 mL Fast plates using an Applied Biosystems QuantStudio S7 (Life Technologies). A single thermal cycling protocol was used for all real-time PCR reactions: 10 min 95 °C, 45× (15 Section 95 °C, 1 min 60 °C). Results were analyzed and exported using the QuantStudio software (version 1.7.2).

**Droplet digital PCR** reactions were performed using the Biorad QX200 digital droplet platform using Twin.Tec 96-well PCR plates (Eppendorf, Aarschot, Belgium). The initial volume of the reaction mixture was 20 μL, which, together with the droplet-generating oil, resulted in a final PCR volume of approximately 45 μL. Reactions were set up using either the Evagreen Supermix (Biorad, Temse, Belgium) or supermix for probes (Biorad), primers and probes ordered from Invitrogen, and nuclease-free water (Ambion). Final primer concentration was 200 nM, with the exception of the probe-based assays (see Table A2 for concentrations). DNA template input varied from 15 to 25 ng per reaction depending on the concentration of the DNA extract. Thermal cycling was performed on a ABI Veriti using the following thermal cycling protocol: 10 min 95 °C, 45× (15 Section 95 °C, 1 min 60 °C), 10 min 98 °C. Results were analyzed and exported using the Quantasoft 1.6.6.320 software.

### 2.5. ddPCR-Based Quantification

The number of a non-declared species in oregano was estimated based on the measurement of the copy numbers of both oregano and the non-declared species by ddPCR. The number of oregano target copies and the number of non-declared species target copies were measured by the relevant assay, corrected for ploidy and the number of genome copies of the target, to obtain the copy number percentage of the non-declared species in the mixture [27].

### 2.6. Sequencing and Metabarcoding

**Barcode PCR amplification.** The five barcodes recommended by the Consortium for the Barcode of Life (CBOL) Plant Working Group [28] were used for metabarcoding. Since the primers targeting the five barcodes have different annealing temperatures, five separated PCR reactions were performed. Usually, 40 ng of DNA was used in each reaction. The barcodes, the primers, and the annealing temperatures are shown in Table 2.

The PCR reaction volume was 50 μL with primers obtained from Invitrogen and using Gold 360 Mastermix (Applied Biosystems, Bleiswijk, The Netherlands), DMSO (Merck, Darmstadt, Germany), and nuclease-free water (Ambion). Thermal cycling was carried out on a GeneAmp PCR system 9700 (Applied Biosystems) using the following protocol: 10 min 95 °C, 35× (30 s. 95 °C, 30 s. (temperature see Table 2), 40 s. 72 °C), 7 min 72 °C. PCR products were separated by agarose gel electrophoresis, purified using a column-based PCR purification kit (PureLink PCR Purification Kit, Invitrogen), and quantified by fluorescence measurements (Qubit, Invitrogen). The purified and quantified amplicons of each sample were pooled together in equimolar quantities. The barcode pools were used as starting material to prepare the DNA barcode libraries for NGS.

**Library preparation and sequencing.** The libraries were prepared using the Ion Plus Fragment Library Kit (Thermo Fisher, Monza, Italy), following the manufacturer’s recommendations [29]. All libraries were evaluated for their quality (expected size range) using an Agilent 2100 Bioanalyzer. Subsequently, the libraries were pooled in an equimolar quantity into the template reaction for the attachment of the fragments to Ion Sphere Particles (ISP) and clonal amplification in emulsion PCR. The template reaction was conducted on the Ion OneTouch 2 instrument (Thermo Fisher, Monza, Italy). Next, recovery and enrichment were performed. Enriched samples were subsequently sequenced on the Ion GeneStudio S5 System (Thermo Fisher), using the Ion 520 chip, which produced 3–5 million reads (1–2 Gb).

### 2.7. Data Processing

**DNA accounting data analysis:** All calculations and curve fittings were conducted using R [30] version 3.5.2 (20 December 2018) ‘Eggshell Igloo’. The data were exported from the droplet reader as ‘csv files’ and imported into R. Droplet calling was performed using the approach presented in [31] using the ‘cloudy’ algorithm version 3.07 as retrieved from Github (https://github.com/Gromgorgel/ddPCR) (accessed on 25 September 2020). Non-NGS sequence analysis (e.g., for primer design and local alignments) was performed in R using functions available through Bioconductor [32] and the ‘DNR’ package available through Github (http://www.github.com/Gromgorgel/R_Scripts) (accessed on 25 September 2020). Online tools and data resources used were the Phytozome database [33], Primer3 [34], Bisearch [35,36], Clustal Omega [37,38,39], in silico PCR [40], the Kew C-value database [41], and Genbank [42].

**NGS data analysis:** The sequencing data obtained were analyzed on the Torrent Suite software and then with a custom-tailored software for species identification (Torrent Suite version 5.16.1), provided by Thermo Fisher. The software clustered all the reads and then BLASTed them against the NCBI nt database (downloaded locally), providing, as results, the number of reads attributed to a species with a certain degree of similarity (by default higher than 99%). In this way, a list of the species detected in each sample was obtained. The results were then analyzed to evaluate how many reads were attributed to the species of interest, and how many reads to possible contaminants or adulterants.

## 3. Results and Discussion

### 3.1. Workflow

After DNA extraction and dilution to a suitable concentration range, all samples were screened for purity of the single-species ingredient (i.e., oregano) with droplet digital PCR using the ‘DNA accounting’ method [27], in which the number of target copies measured by PCR is compared to the ‘expected’ number of target copies calculated from its fluorometrically measured DNA concentration. Out of the 285 samples, 161 samples (56%) had a copy number measured below the lower limit of the expected range, whereas 124 samples (44%) had results within the bounds of the expected range (see also [43]). All samples were analyzed by metabarcoding to investigate its potential as a tool for detecting food fraud. In case the results of the NGS analysis indicated the presence of adulterants, a copy number-based percentage was calculated using ddPCR (if a specific assay was available).

### 3.2. Metabarcoding by NGS

Metabarcoding is an extremely sensitive technique capable of detecting even traces of exogenous species in an otherwise pure sample. However, depending on the DNA quality of all species present in a sample, the ability of the extraction process to recover DNA with similar PCR efficiency from all species, the amplification bias of several PCR reactions involved in the workflow, and the inevitability of small sequencing errors, there is a certain probability of either missing species that are present in a sample or incorrectly identifying them. These same factors contribute to the uncertainty on the quantitative aspect of NGS (i.e., using the read composition of an NGS result as a proxy for the biological composition of a sample).

This is illustrated by Figure 1, which shows metabarcoding results of three oregano plant voucher specimens obtained from Meise Botanical Garden, Belgium. In all three samples, two-thirds of the reads were attributed to *O. vulgare* and around 10% to *Origanum* spp. Thus, in total, 70–82% of sequencing reads belonged to the *Origanum* genus, with the remainder of the reads spread over a limited number of species. *Thymus vulgaris*, *Mentha x piperita*, and *Salvia/Perilla* reads were found in all three samples at proportions of 2–11%. As *Origanum*, *Thymus*, *Mentha*, and *Salvia* all belong to the *Mentheae* tribe of the *Lamiaceae* family, their phylogenetic proximity and small sequencing errors could explain the obtained metabarcoding results. Therefore, the identified members of the *Mentheae* tribe contributed to 89%, 98%, and 97% of the reads for the three oregano vouchers. In one voucher, *Convolvulus* spp. (3%) and *Camonea* spp. (2%), typical field weeds both belonging to the *Convolvulaceae* family, were reported as well.

To assess the capabilities of the metabarcoding approach for detecting extraneous material, leaves from commercially available *Origanum vulgare* and *Olea europaea* plants were dried, milled, and used to gravimetrically prepare mixtures with 1%, 2%, and 5% (*m/m*) olive leaves in oregano. These were then subjected to extraction, barcode amplification, and sequencing.

The results obtained show that the metabarcoding approach is able to detect the presence of olive DNA in all gravimetrically prepared mixtures. However, the percentage of reads did not directly correspond to the adulterant mass percentage: 0.9%, 0.98%, and 1.24% of the total reads were attributed to *Olea europaea* for the 1%, 2%, and 5% by mass, respectively.

The metabarcoding approach also detected and identified all species in the ESA quality control material, except sumac (Table 3). The latter might be due to an inability to extract sufficient DNA from the sumac in the sample; in our laboratory, sumac had a consistent very low yield during DNA extraction (<0.5 ng/μL).

These results illustrate that metabarcoding results reflect not only the presence of adulterants, but also the contact the sample has had with another biological material along the value chain. Depending on the barcodes used, this can range from bacteria, fungi, weeds, insects, etc., present in the agricultural environment, to other spices processed or packaged in the same factory, and eventually to the DNA of the people handling the products during their production.

In all 285 commercial samples analyzed, *O. vulgare* was detected and, as expected from the analyses of the vouchers from Meise Botanical Garden, accompanied by reads from *Origanum onites*, *Origanum* spp., *Mentha x Piperita*, *Salvia/Perilla*, and *Thymus vulgaris* (Figure 2). Weeds were reported in a large proportion of samples, e.g., 84% for *Convolvulus* spp., 38% for *Camonea* spp., 31% for *Ipomoea* spp.

Members of the *Convolvulaceae* (e.g., bindweed or morning glory) family are among the most problematic weeds in agricultural fields, and this may explain their identification as NGS reads as a result of field contamination. In addition, the presence of exogenous DNA could be the result of wind-borne pollen or cross-contact during processing. However, *Olea europaea*, which cannot be considered a weed, was reported by the metabarcoding analysis in 27% of samples.

Another source of unexpected species/genus reads could be the fact that NGS bioinformatics pipelines do not require an exact match to attribute a read to a species, but rather require a sequence similarity higher than a certain threshold value (in this study 99%). However, many species are so closely related that sequencing errors of 1–2 bases may change the species attribution. For short reads, this change can cover quite a large distance in the phylogenetic tree. As such, there is a certain base level of ‘noise’ (species that are reported but are not truly present) in a sample. This is where the read count plays an important role in the data interpretation: very low read counts (i.e., read% ≤ 5%) of species foreign to the production area of oregano are an indication that the species attribution could be wrong.

The raw pipeline output of the 285 oregano samples was therefore filtered by the phylogenetic kingdom and limited to ‘*Plantae*’ (fungi, bacteria, animals, etc., were removed from the list). After filtering, around 90 plant species and families remained (see Table 4, see also [44]).

Initial classifications comprised ‘ingredients’ (*Origanum* spp., but excluding *Origanum majorana*), ‘noise’ (incorrect attributions, either rare or geographically unlikely), ‘contaminants’ (plants or spices with a higher trade value than oregano and agricultural contaminants such as weeds and volunteer plants), and ‘adulterants’ (bulking agents and substitutes reported in the literature). Samples with reads for the latter category were always subjected to ddPCR confirmation of the presence of the relevant species. In the case of ‘contaminants’, their presence was confirmed by ddPCR only in cases of elevated read counts (i.e., read% > 5%).

Among the adulterants, the presence of *Olea europea* and *Myrtus communis* was most often reported in the NGS results (78 and 47 out of 285, respectively), with *Cistus* spp. only reported in five samples. Among the ‘contaminants with high read count’, *Convolvulus* stood out as often reported (238/285) with read percentages up to 72%, whereas *Chenopodium* was reported in fewer samples (45/285) but with read percentages up to 55%.

### 3.3. PCR-Based Quantification

In total, 158 samples were selected for quantification using digital droplet PCR. Most of these were samples in which adulterants were found by metabarcoding, the remainder were a selection of samples with elevated contaminant reads. Table A1 of Appendix A lists the PCR quantification results for these samples as well as the read percentages reported by the initial sequencing analysis.

Figure 3 presents a comparison between the ddPCR quantification results and the NGS read percentages for the contaminants. The correlation between both approaches is strongly species-dependent (see Table 5). This could be explained by differences in DNA extractability, error accumulation during the PCR amplification steps of the different barcodes used, NGS error rate, etc.

The NGS read % of bindweed (*Convolvulus arvensis*) had no functional relationship with the PCR-based copy %. A similar observation was made for white goosefoot (*Chenopodium album*), despite the significant correlation coefficient; in fact, NGS read % was functionally related to copy % (r = 0.82), but the slope of the linear regression function was low, meaning that there was a severe and systematic over-reporting of *C. album* among the reads. Of the 28 samples analyzed, only 2 samples had an estimated contaminant content higher than 2%, with most samples having an average weed content of around 0.75%.

Olive leaf (*Olea europaea*), the most common adulterant in our results, was found in 78 samples, and often (71%) had elevated ddPCR quantification results (>5%). Overall, there was a stronger correlation between the number of *Olea europaea* reads found in oregano samples and its actual olive leaf content than for bindweed and white goosefoot. Samples with an elevated NGS read % (>5%) also showed an elevated copy %, as measured by ddPCR (r = 0.68). However, using the NGS read % as an indicator of the magnitude of adulteration with olive plant material can still be misleading. In nearly all cases, the presence of *Olea europaea* was under-reported by the metabarcoding compositional analysis (see also Figure 3). The results for *Cistus* and *Myrtus* resemble those of olive leaf: samples with more reads most often show higher values in PCR quantification, but with a consistent under-reporting in the number of reads compared to the measured presence.

These results are also reflected in the analysis of the ESA quality control material (see Table 3): over-reporting of *Convolvulus* spp. (27% reported, 1% mass fraction) and under-reporting of *Olea europaea*, *Cistus*, and *Myrtus* (respectively, <1%, 4%, <1% reported, 10%, 10%, 10% mass fraction).

## 4. Conclusions

An inventory made by researchers from Wageningen University and Research places herbs and spices at the top of nine products most vulnerable to adulteration [47].

French authorities (Direction générale de la concurrence, de la consommation et de la répression des fraudes) investigated, in 2019, anomalies in the domestic spice market, and found irregularities in 26.4% of the 138 samples analyzed (cumin, curcuma, paprika/chilli, oregano, pepper, saffron). In an earlier investigation, carried out in 2016, the suspicion rate was 50% [48]. Oregano was frequently reported to be adulterated with other botanicals (olive leaves, myrtle leaves, sumac leaves, cistus leaves, hazelnut leaves) of lower economic value. Alongside conventional wet chemistry methods described by the ISO and relevant trade associations for assessing marketing quality characteristics, such as volatile oil and ash content, chemical profiling of essential oil by GLC [49] or more advanced chromatographic and spectroscopic methods for detecting adulterated oregano are available [5,6,7,50,51]. The disadvantage of such methods is the need for comprehensive reference samples of known identity for building chemometric classification models.

DNA sequence information of a wide range of plants, including culinary herbs, is publicly accessible, and can be used for designing assays to assess oregano authenticity, either based on RAPD [52] or SCAR [46] markers, or by barcoding in combination with qPCR [53]. Such methods target specific adulterants, and designing assays and testing for tens or hundreds of potentially present species in a complex food matrix is not a practical approach. Metabarcoding offers a solution, as this approach does not require a priori information regarding which species shall be targeted. However, metabarcoding still has limitations, such as DNA fragmentation, low DNA yield (particularly from material present at low levels), DNA amplification biases, PCR chimeras (PCR chimeras: artifacts originating from multiple targets, e.g., when the PCR product of one target functions as a primer for a different region in the next PCR cycle, resulting in a sequence that is a composite of multiple targets), primer universality of barcodes, DNA amplification inhibitors, misidentification due to sequence homology, database sequence misannotation, and accidental contamination during sample preparation and analysis [13,54].

The metabarcoding analysis of oregano voucher specimens did not provide the expected answer: in addition to *O. vulgare* and *Origanum* spp., species belonging to the *Mentheae* tribe, i.e., *Mentha x Piperita*, *Salvia/Perilla*, and *Thymus vulgaris*, were reported as well. The close phylogenetic relationship among them may have led to misannotation, resulting in the apparent presence of exogenous material in the vouchers. Consequently, the low read % of these specific plants, as found by NGS, in commercial samples meant that they were not interpreted as contaminants or adulterants.

We used a set of barcodes recommended by the Consortium for the Barcode of Life (CBOL) Plant Working Group, i.e., RbcL, TrnL, psbA, MatK, and ITS, for metabarcoding by NGS of plants. The correct identification of all species in a set of the quality control sample provided by the European Spice Association demonstrated the effectiveness of the approach (see Table 3). All botanicals, except sumac, were correctly identified; however, not all of them to the species level.

This study reports the outcome of one of the largest cross-sectional surveys of the EU market for oregano. All oregano samples included in the study were declared as single-ingredient products. While in all samples, oregano (*O. vulgare*, *Origanum* spp.) was reported by our analysis, the vast majority of samples (>95%) contained reads of other species as well (see Figure 2). In another study of commercial oregano samples [53], the authors reported samples with a total absence of *O. vulgare* reads and a marked presence of *Satureja pilosa/S. montana* (winter savory or mountain savory), which they attributed to similarity between the trnL regions of these species. However, no samples without oregano reads were found in this study, and winter/mountain savory reads were only observed sporadically.

Several groups have already used metabarcoding by NGS for checking the purity of oregano, and the reported results are in good agreement with this study: Barbosa et al. [17], without detailing the barcodes, found ten out of ten oregano samples contaminated with either *Convolvulus arvensis*, regarded as a field contaminant, and/or *Origanum majorana/Origanum onites/Origanum syriacum*, regarded as field or processing contaminants. As described in another study [18], which sampled oregano from the Norwegian market and employed the internal transcribed spacer nrITS2 as the barcode, 23 oregano samples contained undeclared plant species. Their presence could be explained through contamination from wind-pollinated or wind-spread species. Identified species included *Thymus* spp., *Mentha longifolia*, *Polygonum* spp., and *Veronica* spp., which are rather similar to the exogenous species we found as well; however, the whole spectrum of identified non-declared plants differed to quite an extent. The different origin of the samples, as well as the difference in the applied barcodes and the bioinformatics pipeline, could be the reason for this discrepancy, as a broader set of barcodes may be better suited for distinguishing related species [55].

In general, the results show that NGS is a powerful tool for the compositional analysis of food samples, since there were very few cases where the presence of a species found through barcoding was not confirmed with species-specific PCR. However, data interpretation of metabarcoding results can be very challenging, as sequencing errors, truncated reads, and the phylogenetic complexity of the plant kingdom may obfuscate the true composition of a sample. In addition, as others have already reported [53,56], the distribution of reads across species in a sample is a very poor predictor of the actual weight-by-weight constitution of the sample.

Our observations are in broad agreement with the findings that metabarcoding is a powerful technique for identifying species that are present in a composite food sample, but may give inaccurate estimates of its species composition [13]. The outcome of 16S rDNA metabarcoding (read %) of meat products reflected, to a remarkable degree, the species composition of binary meat mixtures for species present at <5% (mass/mass), but larger deviations from the true mixture composition were seen for sausages made from several meat species [57]. There are many probable reasons for the poor correlation between NGS read % and the actual composition of a biological material: barcodes are often located on non-nuclear genes (e.g., located on the mitochondrial or chloroplast genome) whose copy number per cell is tissue-dependent. In addition, the metabarcoding process involves several PCR steps, each of which may show preferential amplification for certain targets, and different barcodes tend to work better for different species. In addition, recovery may not always be equal for all targets throughout the purification and enrichment steps. The addition of errors and uncertainties eventually creates a biased representation of the sample composition (see also [13,58,59,60], and references therein). The results presented in this paper indicate that, although in most cases all species truly present in a biological material were correctly identified by metabarcoding, they were seldom represented proportionally by read %. Therefore, taking the NGS results, in particular, the read counts, at face value could be strongly misleading when deciding whether the presence of DNA of extraneous species is the result of contamination due to inadvertent cross-contact or if it is a possible fraud case (i.e., the addition of bulking agents). Therefore, the use of (semi)-quantitative methods for establishing the level of contaminants, including those based on PCR, is necessary to come to correct conclusions.

In addition, this paper introduces several new sets of primers for the detection and quantification of contaminants and adulterants in oregano samples. These primers were designed to be used with standard reaction conditions and protocols, allowing for quick adoption by other laboratories, thereby contributing to the improved control of herbs and spices to better protect honest business operators and consumers.

## Figures and Tables

**Figure 1 foods-12-02998-f001:**
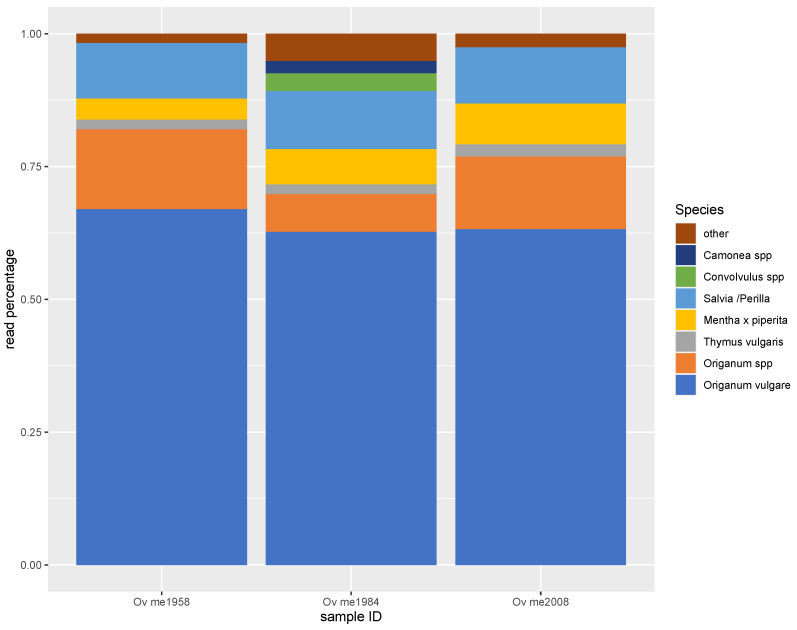
Metabarcoding read distribution of three oregano plant vouchers from Meise Botanical Garden, Belgium.

**Figure 2 foods-12-02998-f002:**
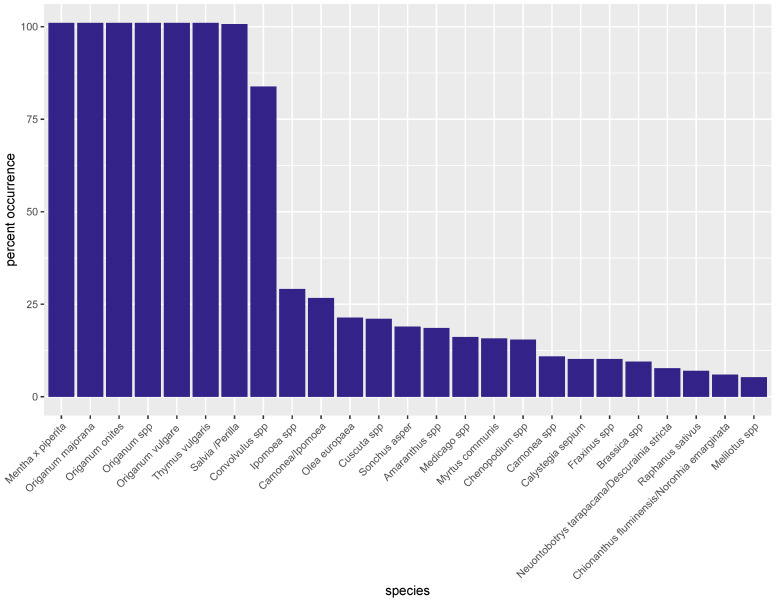
Overview of the species identified in oregano commercial samples (*n* = 285) as measured by metabarcoding. The y-axis shows the percentage of samples in which the plant species (x-axis) was found (at least one read). Species present in less than 5% of samples are not shown.

**Figure 3 foods-12-02998-f003:**
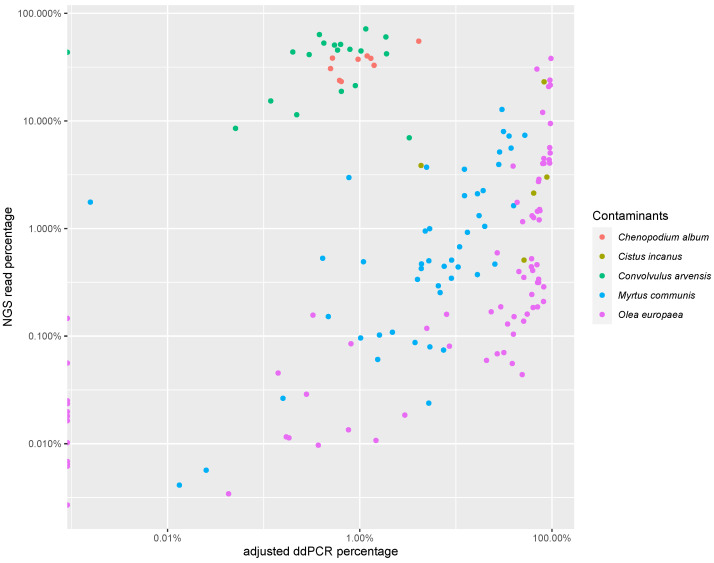
Overview of the quantification results. Both axes are in log scale to adequately represent the orders of magnitude across which the results are spread. The figure shows the relation between the percentage of reads attributed to a contaminant and its measurement in ddPCR.

**Table 1 foods-12-02998-t001:** Sequences and amplicon lengths of primers used in this study. The ‘target’ column gives the Genbank accession number of the sequences from which the primers were designed (primers from this study), or from which the amplicon lengths were estimated (primers published elsewhere).

Name	Forward (5${}^{\prime }$-3${}^{\prime }$)	Reverse (5${}^{\prime }$-3${}^{\prime }$)	Length	Reference	Target
Orvu EF1 F/R	CTCCAGTTCTTGATTGCCACAC	GCTCCTTTCCAGACCTCCTATC	87	[23]	GU385981.1
Oleu SADbis F/R	ATTTCTCATGGAAACACGGC	TTTCATGGCGCTTCTCATC	100	This study	KX196198.1
Chal pc-1E1p1 F/R	AGGACTACCACTGAATCTGC	CTCCAAATCCAAGCCACACA	193	This study	KJ161681.1
Coar HSSp2 F/R	CCCGGTCTAATCGTTGACAT	CAAGGATAAGCGCTCCAGTC	174	This study	HF911513.1
Myrtus isoprene F/R	GTCCATTGAAGGTTACAGCC	CTCCATTAGTCTATCCCTCG	171	This study	FR692046.1
Cistus S13593 F/R	GCGGAAAACCAACAAACCAC	CTACCAATCCTTCCGAACCA	176	This study	AF492022.1

**Table 2 foods-12-02998-t002:** List of barcodes with primer sequences, annealing temperature, and the mean expected amplicon size.

Barcode Name	Primer Name	Sequence (5${}^{\prime }$-3${}^{\prime }$)	Annealing Temp	Amplicon (bp)
RbcL	rbcL-a-F	ATGTCACCACAAACAGAGACTAAAGC	55 °C	560
	rbcL-a-R	GTAAAATCAAGTCCACCRCG		
TrnL	trnL(UAA)-c	CGAAATCGGTAGACGCTACG	50 °C	500
	trnL(UAA)-d	GGGGATAGAGGGACTTGAAC		
psbA	psbA-trnH –F	GTTATGCATGAACGTAATGCTC	64 °C	430
	psbA-trnH-R	CGCGCATGGTGGATTCACAATCC		
MatK	matK-1RKIM-F	ACCCAGTCCATCTGGAAATCTTGGTTC	52 °C	800
	matK-3FKIM-R	CGTACAGTACTTTTGTGTTTACGAG		
ITS	ITS2-F	ATGCGATACTTGGTGTGAAT	56 °C	460
	ITS2-R	GACGCTTCTCCAGACTACAAT		

**Table 3 foods-12-02998-t003:** Declared composition of the European Spice Association (ESA) quality control material (upper section of the table) and the attributed botanicals identified within it by metabarcoding. The lower section of the table lists other genera/species that were found by metabarcoding but not included in the declared composition. The second and third columns show the number of reads as absolute values and as percentages of total reads. The last column lists the declared composition of the control material as mass fraction.

	Reads	NGS Reads%	Declared Mass%
*Origanum vulgare*	1126	8%	22%
*Origanum onites*	109	1%	22%
*Origanum* spp.	564	4%	
*Thymus* spp.	345	2%	5%
*Convolvulus* spp.	3771	27%	1%
*Cistus* spp.	606	4%	10%
*Myrtus communis*	25	<1%	10%
*Olea europaea*	33	<1%	10%
*Corylus* spp.	531	4%	10%
*Rhus coriaria*	-	-	10%
*Amaranthus* spp.	1105	8%	-
*Camonea/Ipomoea* spp.	505	4%	-
*Calystegia* spp.	2974	21%	-
*Chenopodium* spp.	1306	9%	-
*O. majorana*	4	<1%	-
*Mentha x piperita*	133	1%	-
*Salvia/perilla* spp.	133	1%	-

**Table 4 foods-12-02998-t004:** Overview of species found in oregano samples, as reported by NGS, and the classes they were attributed to. For plants to be classified as ‘contaminants’, they should be either common weeds (e.g., *Chenopodium* spp.) or spices/nuts/foodstuffs likely to be handled along the production chain of oregano (e.g., thyme). For plants to be classified as ‘noise’, they should be rare or geographically unlikely (e.g., *Panax stipuleanatus* is an endangered plant endemic to China). For plants to be classified as ‘adulterant’, they should have previously been reported in the literature as adulterant in oregano [7,45,46].

Species	Class	Species	Class	Species	Class
*Origanum majorana*	Adulterant	*Conyza* spp.	Contaminant	*Olea europaea*	Adulterant
*Origanum onites*	Ingredient	*Corylus* spp.	Contaminant	*Panax stipuleanatus*	Noise
*Origanum vulgare*	Ingredient	*Cuminum cyminum*	Contaminant	*Perilla* spp.	Noise
*Aloysia* spp.	Noise	*Cuscuta* spp.	Noise	*Petroselinum crispum*	Contaminant
*Alyssum* spp.	Contaminant	*Cuscuta japonica*	Noise	*Plantago* spp.	Contaminant
*Amaranthus* spp.	Contaminant	*Daucus* spp.	Contaminant	*Raphanus sativus*	Contaminant
*Aniba hostmanniana*	Noise	*Descurainia sophia*	Contaminant	*Reseda lutea*	Contaminant
*Anisosciadium* spp.	Noise	*Descurainia stricta*	Contaminant	*Rhodamnia argentea*	Noise
*Anisosciadium lanatum*	Noise	*Ephedra alata*	Noise	*Rhodostemonodaphne rufovirgata*	Noise
*Arbutus* spp.	Contaminant	*Erigeron* spp.	Noise	*Salvia* spp.	Noise
*Artemisia* spp.	Contaminant	*Erysimum* spp.	Noise	*Saposhnikovia divaricata*	Noise
*Atriplex* spp.	Contaminant	*Erysimum teretifolium*	Noise	*Satureja* spp.	Contaminant
*Avena* spp.	Contaminant	*Fraxinus* spp.	Noise	*Sinocrassula yunnanensis*	Noise
*Bidens* spp.	Contaminant	*Galinsoga parviflora*	Contaminant	*Solanum* spp.	Contaminant
*Brassica* spp.	Contaminant	*Helianthemum* spp.	Contaminant	*Sonchus asper*	Contaminant
*Calycolpus* spp.	Noise	*Hypericum* spp.	Contaminant	*Sonchus* spp.	Contaminant
*Calycolpus moritzianus*	Noise	*Ipomea* spp.	Contaminant	*Syringa* spp.	Noise
*Calystegia sepium*	Contaminant	*Laurus nobilis*	Contaminant	*Syringa wolfii*	Noise
*Camelina* spp.	Contaminant	*Malva* spp.	Contaminant	*Tessaria* spp.	Noise
*Camonea* spp.	Contaminant	*Malva parviflora*	Contaminant	*Thymus* spp.	Contaminant
*Camonea umbrellata*	Contaminant	*Medicago sativa*	Contaminant	*Thymus vulgaris*	Contaminant
*Carpinus viminea*	Noise	*Medicago* spp.	Contaminant	*Thymus marschallianus*	Contaminant
*Carthamus tinctorius*	Contaminant	*Melilotus albus*	Contaminant	*Trifolium* spp.	Contaminant
*Chenopodium album*	Contaminant	*Melilotus officinalis*	Contaminant	*Trigonella* spp.	Contaminant
*Chenopodium* spp.	Contaminant	*Melilotus* spp.	Contaminant	*Valerianella* spp.	Contaminant
*Chionanthus* spp.	Noise	*Mentha x piperita*	Contaminant	*Vicia narbonensis*	Contaminant
*Cicer arietinum*	Contaminant	*Mentheae* (tribe)	Contaminant	*Vicia sativa*	Contaminant
*Cinnamomum* spp.	Contaminant	*Myrcia sylvatica*	Noise	*Vicia* spp.	Contaminant
*Cistus* spp.	Adulterant	*Myrtus communis*	Adulterant		
*Convolvulus arvensis*	Contaminant	*Nama undulata*	Noise		
*Convolvulus* spp.	Contaminant	*Neuontobotrys* *tarapacana*	Noise		

**Table 5 foods-12-02998-t005:** The *p* values, correlation coefficients, and slopes of the linear regressions for the data in Figure 3.

	*C. album*	*Cistus* spp.	*C. arvensis*	*M. communis*	*O. europaea*
n	9	5	19	47	78
Correlation coefficient	0.82	0.39	−0.05	0.71	0.68
*p* value	*p* < 0.001	*p* > 0.1	*p* > 0.1	*p* < 0.001	*p* < 0.001
Slope	0.09	0.65	0	4.6	1.96

## Data Availability

The data used to support the findings of this study can be made available by the corresponding author upon request.

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
