# Peer review of "Detection and Quantification of Botanical Impurities in Commercial Oregano (*Origanum vulgare*) Using Metabarcoding and Digital PCR"

_foods, 2023, doi:10.3390/foods12162998_

Round 1

Reviewer 1 Report

The manuscript submitted by Antoon Lievens et al. deals with the authentication of commercial oregano with respect to the presence of botanical impurities. The study investigated the applicability of DNA metabarcoding for the detection and quantification of adulterants and contaminants in 285 oregano samples.

The topic of the study is very interesting and the scientific merit of the manuscript is sound. However, some revision is necessary.

 1)     I suggest citing one or two references in the first paragraph of the introduction.

2)     For a better unterstanding, I would not use different terms for one and the same technology. In the title, it is called MetaBarcoding, in the fourth paragraph of the introduction section metabarcoding, in the fifth paragraph of the introduction section NGS metabarcoding, and in the legend of Figure 1, DNA Metabarcoding. For readers that are not familiar with the technology, this might be confusing.

3)     Ad experimental section: Most probably, the experimental section including Table 2 was generated by copying paragraphs from a previous paper published by the authors (Lievens, et al.  DNA Accounting: Tallying Genomes to Detect Adulterated Saffron. Foods 2021, 10, 2670. https://doi.org/10.3390/foods10112670.) I suggest rephrasing the experimental section.

4)     The title of section 2.4. is ddqPCR. I guess it should be ddPCR.

5)     Ad section 2.5, first paragraph of “Since the five barcodes have different annealing temperatures,…” should be replaced by “Since the primers targeting the five barcodes have different…” or “Since the assays for the five barcodes have…”

6)     Ad 2.6., first paragraph: the reference for Kew C-value database is missing.

7)     Ad 3.1. The authors mention that “all samples were screened for purity of the single-species ingredient (i.e. oregano) with droplet digital PCR using the ‘DNA accounting’ method.” In my opinion the authors should give the number of samples for which the number of target copies determined by ddPCR differed from the ‘expected’ number calculated from the Qubit results.

8)     Ad 3.1.  “In case the results of the NGS analysis indicated the presence of adulterants, they were quantified using a calibration curve-based qPCR approach”. The authors should describe “the calibration curve-based qPCR approach” in more detail, including the calibrants that were used. For me it is difficult to understand which of the results presented were obtained by qPCR.

9)     Ad 3.1. As I understand, all samples were analysed by DNA metabarcoding by using the five primer pairs listed in Table 2, targeting five DNA barcodes. For me it is not clear if the results presented, e.g. in Figure 1, Figure 2, Figure 3, Table 4, and Table A1 refer to one of the DNA barcodes and if yes, to which of the five DNA barcodes. In my opinion, it would be interesting to learn which species could be detected with which of the five DNA barcode(s).

10)  Ad Figure 2, y-axis: “occurrance” should be replaced by “occurrence”

11)  Ad Figure 3, legend: “The figure shows the relation between … and its measurement in real time PCR.” I guess that” real time PCR” should be replaced by “ddPCR”.

12)  Page 11, line 11: delete “characteristics” in “…..charactristicscharacteristics….”

13)  Ad Conclusion, fourth paragraph: The term “PCR chimeras” should be explained, in my opinion, it is not very common.

14)  In the abbreviation list, “digital droplet PCR” should be changed to “droplet digital PCR”. The abbreviation “MDPI” can be deleted.

15)  Table A2, legend and column 4: “flourophore” should be changed to “fluorophore”

 Author Response

Dear, thank you for your comments and suggestions. Please find below our responses to your questions.

1)     I suggest citing one or two references in the first paragraph of the introduction.

Relevant references have been added.

2)     For a better unterstanding, I would not use different terms for one and the same technology. In the title, it is called MetaBarcoding, in the fourth paragraph of the introduction section metabarcoding, in the fifth paragraph of the introduction section NGS metabarcoding, and in the legend of Figure 1, DNA Metabarcoding. For readers that are not familiar with the technology, this might be confusing.

The mentions of the technology have been harmonized. It is now referred to as “metabarcoding” or “metabarcoding by NGS”.

3)     Ad experimental section: Most probably, the experimental section including Table 2 was generated by copying paragraphs from a previous paper published by the authors (Lievens, et al.  DNA Accounting: Tallying Genomes to Detect Adulterated Saffron. Foods 2021, 10, 2670. https://doi.org/10.3390/foods10112670.) I suggest rephrasing the experimental section.

Avid readers of our papers might indeed recognize certain phrases as much of the experimental work is similar between these two papers. We have made some minor changes to better distinguish both texts.

4)     The title of section 2.4. is ddqPCR. I guess it should be ddPCR.

This has been corrected.

5)     Ad section 2.5, first paragraph of “Since the five barcodes have different annealing temperatures,…” should be replaced by “Since the primers targeting the five barcodes have different…” or “Since the assays for the five barcodes have…”

This has been corrected.

6)     Ad 2.6., first paragraph: the reference for Kew C-value database is missing.

The reference has been added.

7)     Ad 3.1. The authors mention that “all samples were screened for purity of the single-species ingredient (i.e. oregano) with droplet digital PCR using the ‘DNA accounting’ method.” In my opinion the authors should give the number of samples for which the number of target copies determined by ddPCR differed from the ‘expected’ number calculated from the Qubit results.

The information has been added together with a reference to the complete report of the Coordinated Control Plan on herbs and spices where these results are laid out in more detail.

8)     Ad 3.1.  “In case the results of the NGS analysis indicated the presence of adulterants, they were quantified using a calibration curve-based qPCR approach”. The authors should describe “the calibration curve-based qPCR approach” in more detail, including the calibrants that were used. For me it is difficult to understand which of the results presented were obtained by qPCR.

We apologize for the confusion. The statement about qPCR quantification is a remnant from a previous draft of this paper. All percentages reported in the manuscript were obtained using digital droplet PCR. We made changes to the text to better reflect this.

9)     Ad 3.1. As I understand, all samples were analysed by DNA metabarcoding by using the five primer pairs listed in Table 2, targeting five DNA barcodes. For me it is not clear if the results presented, e.g. in Figure 1, Figure 2, Figure 3, Table 4, and Table A1 refer to one of the DNA barcodes and if yes, to which of the five DNA barcodes. In my opinion, it would be interesting to learn which species could be detected with which of the five DNA barcode(s).

All NGS analyses were performed using 5 plant barcodes together, or as many barcodes as it was possible to amplify by PCR. We have added a statement to section 2.5 explicitly mentioning the pooling of the barcodes since it was indeed not clearly mentioned that all barcodes were used together.

An overview of the success-rate of each barcode with respect to the species detected would indeed be an interesting addition. However, due to the summer holidays and time constraints within which we are required to provide a corrected version we were not able to include this in the manuscript.

10)  Ad Figure 2, y-axis: “occurrance” should be replaced by”d “occurrence”

The figure labels have been updated

11)  Ad Figure 3, legend: “The figure shows the relation between … and its measurement in real time PCR.” I guess that” real time PCR” should be replaced by “ddPCR”.

This is indeed an error, it has been corrected

12)  Page 11, line 11: delete “characteristics” in “…..charactristicscharacteristics….”

This has been corrected.

13)  Ad Conclusion, fourth paragraph: The term “PCR chimeras” should be explained, in my opinion, it is not very common.

A end note with a brief explanation has been added to the manuscript

14)  In the abbreviation list, “digital droplet PCR” should be changed to “droplet digital PCR”. The abbreviation “MDPI” can be deleted.

These changes have been made.

15)  Table A2, legend and column 4: “flourophore” should be changed to “fluorophore”

The typo has been corrected.

Reviewer 2 Report

The report is of very high quality and clearly presents the objectives of the study and details of the methods used and the results obtained. The results of the study and their reliability are discussed in detail and the main findings and their implications are presented in an excellent way.

Here are some comments to improve the paper:

 Editorial Comments:

2.2. PCR Primers

·        Olive tree: correct the spelling: specific primers

·        Myrtle: add “gene”: isoprene synthase gene

·        Cistus: add “gene”: (GGPPS1) gene

2.3 PCR methods

               Droplet digital PCR: change the word expection to exception

2.6 Data processing

               DNA Accounting data analysis: add reference for the Kew C-value database

3.2 NGS Metabarcoding

               .. the ability of the extraction process to recover (instead of recovery) DNA …

4. Conclusions

Paragraph 5, sentence 2: modify to: The correct identification of all species in the quality control sample provided by …

Paragraph 6 sentence 3 modify to: While in all of them oregano (…) was reported …

Paragraph 9 sentence 3: “The results presented in this paper …” – delete “but”

References

               26. First author is missing

40. Bejar, E. 

Technical Comments:

3.2 NGS Metabarcoding, chapter below Table 3

Explain, why sumac was not identified by the metabarcoding method.  Is there any possibility to modify the metabarcoding primers or include further primers/barcodes? According to other investigations Oreganum was reported to be adulterated with sumac leaves.

The ESA quality control sample is a very valuable material not only for the qualitative evaluation of the metabarcoding method but also for showing the limits of NGS quantification Unfortunately these results are not described although the composition of the sample is known. Please add the NGS read-% for this sample and discuss the results.

Author Response

Dear,

thank you for your comments and suggestions. Please, find below our responses to your questions.

Editorial Comments:

2.2. PCR Primers

  • Olive tree: correct the spelling: specificprimers
  • Myrtle: add “gene”: isoprene synthase gene
  • Cistus: add “gene”: (GGPPS1) gene

These changes have been made.

2.3 PCR methods

               Droplet digital PCR: change the word expection to exception

The typo has been corrected.

2.6 Data processing

               DNA Accounting data analysis: add reference for the Kew C-value database

The reference has been added.

3.2 NGS Metabarcoding

               .. the ability of the extraction process to recover (instead of recovery) DNA …

This has been changed.

  1. Conclusions

Paragraph 5, sentence 2: modify to: The correct identification of all species in the quality control sample provided by …

Paragraph 6 sentence 3 modify to: While in all of them oregano (…) was reported …

Paragraph 9 sentence 3: “The results presented in this paper …” – delete “but”

These changes have been made.

References

  1. First author is missing
  2. Bejar, E.

The references have been updated 

Technical Comments:

3.2 NGS Metabarcoding, chapter below Table 3

Explain, why sumac was not identified by the metabarcoding method.  Is there any possibility to modify the metabarcoding primers or include further primers/barcodes? According to other investigations Oreganum was reported to be adulterated with sumac leaves.

We have had difficulties extracting DNA from the sumac material that was available to us. We do not know if that material is comparable to the material used in the ESA sample. It is therefore, uncertain that low extraction efficiency is the cause of the result, but it remains our most likely hypothesis. A sentence suggesting extraction difficulties has been added.

The ESA quality control sample is a very valuable material not only for the qualitative evaluation of the metabarcoding method but also for showing the limits of NGS quantification Unfortunately these results are not described although the composition of the sample is known. Please add the NGS read-% for this sample and discuss the results.

Table 3 was changed to also present the quantitative data from the NGS analysis. As the results reflect our findings from the commercial samples, no in-depth discussion was deemed necessary. Some text was added the end of section 3.3  to highlight the quantitative results.

Reviewer 3 Report

There are a handful of typos; otherwise language is fine.

Author Response

Dear,
thank you for your comments.
All typos that were also pointed out by other reviewers have been corrected.
This is the list of corrected typos:

  1. The title of section 2.4. is ddPCR.
  2. Figure 2, y-axis: “occurrance” replaced by“occurrence”

  3. Page 11, line 11: deleted “characteristics” in “…..charactristicscharacteristics….”

  4. Table A2, legend and column 4: “flourophore” changed to “fluorophore"
  5. 2.2. PCR Primers, Olive tree: spelling corrected: specific primers

  6. 2.3 PCR methods: change the word expection to exception

Moreover, please find the answers to your questions below:

  1. All specificity results were obtained using SYBRgreen chemistry. SYBR green was originally chosen during the coordinated control plan as flexible chemistry that allowed us to design and test primers ‘on the fly’ in a short time-loop when NGS indicated the presence of species for which we had no assays available. The order-times for probe based chemistries did not allow that. Afterwards, we started the transition to probe-based chemistries for the ability to multiplex assays and for the improved specificity of a three-molecular system. We have added text to the table legends to clarify which chemistry was used.
  2.  "undetermined" is related to the Cq values that do not exceed the Cq threshold and, therefore, cannot be quantified. It means Cq>40. A note to clarify this has been added in the table legend.

  3. The original column order was chosen because the results are sorted by contaminant species for ease of interpretation. ‘Sample’ and ‘species’ columns have been switched while keeping the order of the results the same.

  4. The text under the Fig. 2 legend says that samples with NGS read for the adulterant were always subjected to ddPCR. In case of ‘contaminants’, their presence was confirmed by ddPCR only in case of elevated read-counts (i.e. read% > 5%). In Figure 3, only adulterants (i.e. Myrtus, Olea and Cistus species) show NGS reads below 5%.

  5. Changes to the figure legend have been made to clarify this.

  6. The design and validation of a new set of universal plant barcodes was outside the scope of our work. The sequence-specificity issues that emerged from our work are presented as a result and starting point for future research. The barcodes chosen and used in this paper (and during the coordinated control plan) were chosen based on recommendations from the Consortium for the Barcode of Life (as is stated in the text) . The selection of barcodes is unfortunately always a trade-off between sequences that are best suited for the research and sequences that are present in the databases for most species, with the intent of ‘casting as wide a net as possible’ more importance was given to the latter.

  7. While we agree that the a deeper dive into the sources of bias in NGS would be a great addition to the paper, the authors feel that this falls outside the scope of the article and is a research topic of itself best presented in a separate paper. However, we have included additional references in the conclusions that may guide the reader to additional publications that contribute on this topic.

  8. For an in-depth validation of the ddPCR quantification approach, we refer the reader to our previous publication (https://doi.org/10.3390/foods10112670 , see reference n° 27 ) where this approach presented and examined in more detail.

  9. The oregano voucher specimens were leaves taken from the living plant collection of Meise Botanical Garden, their correct identification is guaranteed. The NGS reads from metabarcoding were analysed with BLAST using the entire NCBI nucleotide database (not only reference DNA sequences). We speculate that some of the DNA sequences in the NCBI database were uploaded with the wrong species or genus name. This misclassification might happen when species are very closely related and, therefore, difficult to identify correctly.

    If just few sequences are misannotated in the database, the result of the BLAST alignment will be affected as those misannotated species will appear as good matches. Later in the conclusion chapter, we also specify that the phylogenetic closeness of the species in the Mentheae tribe, together with the presence of truncated reads and sequencing errors in the database might be the cause of misannotation, and consequently, results in wrong sample composition when analysing pure vouchered samples.
